# Broadly Antiviral Activities of TAP1 through Activating the TBK1-IRF3-Mediated Type I Interferon Production

**DOI:** 10.3390/ijms22094668

**Published:** 2021-04-28

**Authors:** Jin Zhao, Ruiting Li, Yanjun Li, Jiaoshan Chen, Fengling Feng, Caijun Sun

**Affiliations:** 1School of Public Health (Shenzhen), Sun Yat-sen University, Guangzhou 514400, China; zhaoj47@mail2.sysu.edu.cn (J.Z.); lirt9@mail2.sysu.edu.cn (R.L.); liyj253@mail2.sysu.edu.cn (Y.L.); chenjsh59@mail2.sysu.edu.cn (J.C.); fengfling@mail.sysu.edu.cn (F.F.); 2Key Laboratory of Tropical Disease Control (Sun Yat-sen University), Ministry of Education, Guangzhou 514400, China

**Keywords:** transporter associated with antigen processing 1 (TAP1), innate immunity, interferon-stimulated genes (ISGs), type I interferons (IFN-Is), broadly antiviral activity

## Abstract

Deeply understanding the virus-host interaction is a prerequisite for developing effective anti-viral strategies. Traditionally, the transporter associated with antigen processing type 1 (TAP1) is critical for antigen presentation to regulate adaptive immunity. However, its role in controlling viral infections through modulating innate immune signaling is not yet fully understood. In the present study, we reported that *TAP1*, as a product of interferon-stimulated genes (ISGs), had broadly antiviral activity against various viruses such as herpes simplex virus 1 (HSV-1), adenoviruses (AdV), vesicular stomatitis virus (VSV), dengue virus (DENV), Zika virus (ZIKV), and influenza virus (PR8) etc. This antiviral activity by TAP1 was further confirmed by series of loss-of-function and gain-of-function experiments. Our further investigation revealed that TAP1 significantly promoted the interferon (IFN)-β production through activating the TANK binding kinase-1 (TBK1) and the interferon regulatory factor 3 (IRF3) signaling transduction. Our work highlighted the broadly anti-viral function of TAP1 by modulating innate immunity, which is independent of its well-known function of antigen presentation. This study will provide insights into developing novel vaccination and immunotherapy strategies against emerging infectious diseases.

## 1. Introduction

Viral infectious diseases have become a serious threat to global public health, and deeply understanding the virus-host interactions is of great importance for developing effective anti-viral strategies. Both innate immunity and adaptive immunity play important roles in controlling viral infections. The interferon (IFN)-mediated innate immunity provides the first line of defense against invading pathogens. Subsequently, adaptive immunity is activated following specific antigen presentation, and the highly specialized and potential humoral immune responses and T lymphocytes-mediated cellular immune responses are induced to further eliminate pathogens or pathogen-infected host cells.

Type I interferons (IFN-Is) are well known for their key roles in controlling viral infections [1,2]. IFN-Is, which include IFNα, -β, and -ω, are rapidly activated during viral infections and then induce a unique set of IFN-stimulated genes (ISGs) [3]. Hundreds of ISGs have been recently reported for their potential antiviral functions by interfering with different stages of viral life cycles. Among them, a transporter associated with antigen processing (TAP) belongs to a member of the superfamily of ATP-binding cassette (ABC) transporters [4]. Traditionally, the TAP complex, which consists of TAP1 and TAP2, can pump the degraded cytosolic antigen peptides across the endoplasmic reticulum into the cell membrane and then bind to the major histocompatibility complex (MHC) class I compartment. Therefore, TAP can play a well-known function of antigen presentation to regulate adaptive immunity [5]. For example, the human papilloma virus (HPV) can evade immune recognition by downregulating the MHC class I -mediated antigen presentation via inhibiting TAP1 activity [6]. In addition, patients with bare lymphocyte syndrome (human genetic TAP deficiency syndrome Type 1) are characterized by down-regulating the level of MHC class I molecules, and usually suffer clinically recurrent infections and necrotizing skin lesions [7], and the pathogenesis for this syndrome is at least partially because MHC class I -mediated antigen presentation plays a critical role in activating the cytotoxic T lymphocytes (CTLs), which can kill the virus-infected cells or the malignant-transformed cells.

Alternatively, studies also demonstrated that TAP played an important impact on regulating innate immunity. TAP1 is ubiquitously expressed in the spleen, appendix, lymph node and other tissues, as well as in many immune cells, including dendritic cells and macrophages. The expression of TAP1 is usually at a low level but significantly upregulated in response to IFN stimulation or viral infections [8,9,10]. Therefore, TAP1 might be a critical factor in the subtle virus-host interactions, especially in innate immune signaling and adaptive immunity. In the present study, we emphasized exploring the broadly antiviral function of TAP1 by modulating innate signaling pathway and further clarified the underlying mechanism, which is independent of its previously well-known function of antigen presentation to regulate adaptive immunity. Importantly, we found that TAP1 significantly promoted the interferon (IFN)-β production through activating TBK1 and IRF3 signaling transduction. These data will be helpful to deeply understand the profound virus-host interactions and provide insights into developing novel vaccination and immunotherapy strategies against a variety of infectious diseases.

## 2. Results

### 2.1. The Elevated TAP1 in Response to Toll-Like Receptors (TLRs) Agonists and Viral Infections

It has been reported that TAP1 played an important role in regulating innate and adaptive immunity, and we began this study to further confirm that TAP1 expression is interferon-dependent. Consistent with previous reports, TAP1 was upregulated dramatically in response to stimulation by multiple kinds of TLR agonists in different macrophage cell lines, including Raw 264.7 cells and mice bone marrow-derived macrophages (J2-BMMs) (Figure 1). Our data showed that the expression level of *TAP1* was significantly upregulated with the stimulation of LPS (Toll-Like Receptor 4 (TLR4) agonist [11]), polyI:C (TLR3 agonist [12]) and R848 (TLR7, TLR8 agonist [13]). Importantly, we also monitored the kinetics of *TAP1* expression at multiple time-points during stimulation, and we found that there was a transient decline of *TAP1* expression at the early several hours upon TLR agonists stimulation and then followed a sharp upregulation (Figure 1A–C). In addition, *TAP1* expression was significantly increased in response to the stimulation with IFN-α and HSV-1 (Figure 1D,E). This upregulation of TAP1 was further confirmed in J2-BMMs by Western blot. Moreover, we found that this upregulation was abrogated in the (*IFNAR−/−*)-J2-BMM cells, confirming that TAP1 expression was IFN-dependent (Figure 1F).

### 2.2. Broadly Antiviral Activities of TAP1

Previous studies showed that TAP1 might play a role against viral infection. To further investigate the breadth of antiviral activity of TAP1, we tested its inhibitory effects on various representative viruses (Figure 2). For example, we detected the effect of *TAP1* on HSV-1 (a kind of enveloped DNA virus) and adenovirus (a kind of nonenveloped DNA virus), and results showed that *TAP1* overexpression significantly inhibited their infection in a dose-dependent manner (Figure 2A,B). In addition, to determine whether *TAP1* can inhibit the replication of RNA viruses, we tested the effect of *TAP1* on VSV and PR8 influenza (a kind of negative-strand RNA virus) (Figure 2C,D), DENV and ZIKV (a kind of positive-stranded RNA virus) (Figure 2E,F), and our results showed that TAP1 significantly inhibited their infections in a dose-dependent manner (Figure 2G–J). We also found that different viruses had various effects on cell viability (Appendix A), which may subsequently affect the level of TAP1 expression. The comparisons of TAP1 levels in uninfected, mock-transfected cells, infected mock-transfected, and the infected and TAP1-transfected cells were showed in Appendix A. Therefore, these data demonstrated that TAP1 can broadly inhibit various DNA viruses and RNA viruses.

### 2.3. Validation of Antiviral Activity of TAP1

Then, using a replication-competent HSV-1 reporter virus, we performed a series of loss-of-function and gain-of-function experiments to further validate the function of viral inhibition by TAP1. As described in Methods, the direct inhibition of viral infection by TAP1 was verified by plaque assay in Vero cells. The amount of plaque-forming units (PFU) of HSV-1 in TAP1-treated samples was significantly dropped than that of negative control samples (pVAX-*GFP*) and similar to that of positive control samples (pVAX-*IRF1*) (Figure 3A). In addition, *TAP1* overexpression in HEK293T cells also significantly inhibited HSV-1 replication through quantifying the expression of TAP1 and HSV-1 *UL27* by RT-qPCR (Figure 3B).

To further confirm the effect of TAP1 on HSV-1 replication, the expression of *TAP1* in Raw 264.7 and WT-J2-BMM cells was down-regulated by small interfering RNA (siRNA), and the knockdown efficiency was determined by RT-qPCR. Our results showed that these *TAP1*-knockdown-cells became more susceptible to HSV-1 infection when compared to mock control (Figure 3C,D). Moreover, we generated the *TAP1* knockout fibroblast cells to further determine the role of TAP1 during HSV-1 infection, and quantification of luciferase activity demonstrated that *TAP1* deficiency (*TAP1*−/− fibroblast cells) significantly enhanced HSV-1 replication than that of *TAP1*+/+ fibroblast cells (Figure 3E). Based on the above loss-of-function experiments, we next performed the gain-of-function experiment to validate the antiviral function of *TAP1*. *TAP1*+/+ or *TAP1*−/− fibroblast cells were transfected with or without *TAP1*-expressing plasmids, followed by HSV-1 infection and then the luciferase activity was detected. Results demonstrated that the replenishment of *TAP1* could effectively restore the antiviral function in *TAP1*−/− fibroblast cells (Figure 3F). We also explored whether TAP1 inhibits the entry of HSV-1. The cells were disposed of as shown at the top of Figure 3G, and the result showed that *TAP1* overexpression inhibited the viral entry, while *TAP1* knockdown promoted the viral entry (Figure 3G). Taken together, the effect on viral inhibition of TAP1 was validated by a series of loss-of-function and gain-of-function experiments.

### 2.4. The Mechanism for Broadly Antiviral Activities of TAP1

Next, we explored the underlying mechanism for the broadly antiviral activity of TAP1. As expected, *TAP1* overexpression significantly induced a high level of *IFN-β* mRNA in HEK293T cells (Figure 4A). In contrast, *TAP1* knockdown could significantly decrease the level of *IFN-β* expression both in WT-J2-BMM and Raw 264.7 cells (Figure 4B,C). In addition, the abundance of IFN-β protein in response to TAP1 stimulation was further validated by ELISA, and we found that the concentration of IFN-β protein was significantly increased in *TAP1*-overexpressing cells while decreased in *TAP1*-knockdown cells (Figure 4D).

We then detected the subsequent signaling pathway involved in TAP1-induced IFN-β production. A previous study demonstrated that the induction of type I interferons might be related to the STING-TBK1-IRF3 signaling axis [14], so we focused on this signaling pathway in the next study. HEK293T, THP-1 cells and Raw 264.7 cells were transfected with *TAP1*-expressing plasmid or mock plasmid, followed by HSV-1 infection, and then the targeted protein was detected by Western blot assay. Consistent with the above data, TAP1 overexpression significantly inhibited the expression of HSV-1 ICP27. Importantly, we found that TAP1 overexpression significantly promoted the phosphorylation of TBK1 and IRF3 (Figure 4E–G). We further demonstrated that TAP1overexpression could activate the TBK1/IRF3 pathway to inhibit DNA viruses (Represent by the ICP27 protein encoded by HSV-1) and RNA viruses (Represent by the NA protein encoded by PR8 influenza) in a dose-dependent manner (Figure 4H,I). In contrast, TAP1 knockdown significantly decreased the phosphorylation of TBK1 and IRF3, and consequently, increased the expression of HSV-1- ICP27 protein in J2-BMM cells (Figure 4J) and HEK293T cells (Figure 4K). Additionally, we further verified that TAP1 replenishment rescued the phosphorylation of TBK1 and IRF3 in the siRNA-treated cells in a dose-dependent manner (Appendix A). Overall, these data indicated that TAP1 affected viral infections by regulating the TBK1/IRF3 signal pathway.

We also observed that the level of *IFN-β* mRNA in the *TAP1* overexpression group was significantly improved in response to various viral infections (Figure 4L), and the elevation of IFN-β protein expression was further confirmed by ELISA (Figure 4M,N). Interestingly, when (*IFNAR−/−*)-J2-BMM cells were stimulated with viral infections, there was a significantly higher level of IFN-β in supernatant than that of cell lysates (Figure 4O), implying that the elevated IFN-β protein cannot enter into cytoplasm normally because of lack of IFN receptor. Of note, SARS-CoV-2 pseudovirus can sharply upregulate the IFN-β expression in different macrophage cell lines (Figure 4M–O), suggesting that TAP1 might a potent inhibitor against SARS-CoV-2 infection through inducing IFN-β production. In addition, the phosphorylation of STAT1, which is a kind of critical protein in the downstream of the IFN-β-activating signal pathway, was blocked in (*IFNAR−/−*)-J2-BMM cells, while the expression level of total STAT1 protein was not affected (Figure 4P). Taken together, TAP1 can broadly inhibit viral infections through activating the TBK1-IRF3 -mediated IFN-β production (Figure 5).

## 3. Discussion

Type I IFNs and their downstream ISGs are well known to play a critical role in host defense by interfering viral life cycle [15,16,17,18,19]. Numerous ISGs have been recently reported, but the underlying mechanism for their potential antiviral functions remains poorly understood. In this study, we investigated the function of TAP1 against broadly viral infections through activating unreported signaling pathways, the TBK1-IRF3-mediated type I interferon production. Previously, several studies reported the function of TAP1, which mainly focused on host defense through regulating adaptive immunity [4,5,6,7,20]. Interestingly, some studies demonstrated that TAP1 negatively regulated the antiviral innate immunity by targeting the TGF-β-activated kinase (TAK)1-TAK1-binding protein (TAB) complex [21]. In addition, previous studies also showed that TAP1 might not play a role in the inhibition of some viruses. For example, TAP1 did not play a significant role in the inhibition of MHV-68, possibly because the viral mK3 protein of MHV-68 can interact with TAP1 and promote its proteasome degradation [22,23]. The exact reason for this discrepancy is not known. Actually, when TAP1 has effectively inhibited viral replication by activating innate immunity, less viral peptide and antigen presentation would be produced, and then adaptive immunity would be decreased [24]. Moreover, it is possible that the different cell lines limit the utilization of certain signaling pathways, and therefore, it would be greater to study the comprehensive impact of TAP1 on virally infected cells in vivo in future work. Of note, evolutionally, TAP1 has been also a target for viral evasive strategies. For example, ICP47 protein encoded by the HSV-1 can bind specifically to the TAP1, thus contributing to virus escaping from immune surveillance [25]. Overall, the multifaced functions of TAP1 might be indeed a good example of profound immune regulatory networks to maintain immune homeostasis.

Another interesting observation in our study is the kinetics of TAP1 expression upon TLRs agonists stimulation, and we reported that there was a transient decline of TAP1 expression at the early several hours before a sharp upregulation. The underlying mechanism for this phenomenon is unknown, but it is consistent with the cell biology of antigen presentation in response to TLR stimulation. TLR signaling can induce phagosomal MHC-I delivery to enhance TAP recruitment to endosomes [26,27], which is beneficial for antigen cross-presentation [28]. The peptide-mediated undocking of MHC class I is linked to the transport cycle of TAP1 [29]. The transient decline of TAP1 expression might be related to the fact that cycling of MHC ceases for a time as MHC-peptide complexes are fixed on cell surfaces to activate T cell-mediated adaptive immunity [30]. Thus, further study can explore whether TAP1-conjuncted antigens mediate the crosstalk between innate immunity and adaptive immunity and subsequently improve the immunogenicity of vaccine candidates.

Taken together our data and previously reported works, we proposed a work model to illustrate how TAP1 plays a critical factor in the profound virus-host interactions, especially in an essential function of antiviral immunity (Figure 5). In brief, TAP1 can promote type I interferon production by activating the phosphorylation of the TBK1-IRF3 signal pathway. Subsequently, Type I IFNs can bind to IFNAR2, and then recruit IFNAR1, and consequently form a signaling-competent ternary complex to activate the heterotrimeric transcription factor complex interferon-stimulated gene factor 3 (ISGF3), which is comprised of phosphorylated signal transducer and activator of transcription (STAT)1 and STAT2 and interferon regulatory factor 9 (IRF9), through the Janus kinase (JAK)-STAT signal pathway [31,32,33]. The activated ISGF3 can translocate to the nucleus and bind to IFN-stimulated response elements (ISREs) in the upstream promoter regions of ISGs [34,35] and then encode the numerous proteins with potent antiviral activities [18,36].

The frequent outbreaks of infectious diseases, especially due to highly pathogenic viruses, have been a severe and persistent challenge for global public health, but there are still no enough effective antiviral strategies. With the long-term competition and mutual evolution, there are profound and subtle interactions between host immune cells and viruses. As a result, it is of great importance to deeply study the complicated virus-host interactions. Previous studies demonstrated that TAP1 could regulate both innate immunity and adaptive immunity, and we focused on TAP1-involved antiviral innate immunity in this study. Previous studies had shown that TAP1 expression could be induced by IFNs or viral infections, such as hepatitis C virus (HCV), Sendai virus (SeV), VSV and influenza A virus et al. [8,9,10,21,37]. Consistent with previous reports, we observed that IFN-β expression was dramatically elevated by TAP1 expression, implying that the inhibition of viral infection by TAP1 is closely associated with the IFN signaling pathway. Meanwhile, it is well known that IFN-I can induce the expression of ISGs (including TAP1) through activating the JAK-STAT pathway in a positive feedback manner [38]. Nevertheless, we further expanded the breadth of antiviral activity of TAP1 in this study. Of note, it is recently reported that IFN-β appeared to be pivotal to ameliorate the disease progression in a combined therapy regiment of IFN-β, lopinavir–ritonavir, and ribavirin at the early stage of the coronavirus disease 2019 (COVID-19) [39,40]; therefore, TAP1 might be a potent inhibitor against SARS-CoV-2 infection. Giving frequent outbreaks of emerging viruses, it is no doubt an urgent need to exploit an effective broad-spectrum antiviral drug.

In conclusion, this study enriched our understanding of the broadly antiviral function of TAP1 by regulating innate immunity and provided insights into developing novel antiviral strategies against a variety of emerging infectious diseases.

## 4. Materials and Methods

### 4.1. Cells and Viruses

#### 4.1.1. Cells

HEK293T cell (from the embryonic kidney of a female human fetus), Vero cells (from the kidney of a female normal adult African green monkey), and RAW 264.7 cells (macrophage of a male adult mouse) were cultured in complete Dulbecco’s modified Eagle’s medium (DMEM, Gibco, Grand Island, NY, USA) containing 10% fetal bovine serum (FBS, Gibco) and 1% penicillin/streptomycin (Gibco), at 37 °C in an atmosphere of 5% of CO_2_.

THP-1 cell (human myeloid leukemia mononuclear cells) was cultured in complete RPMI 1640 medium containing 10% fetal bovine serum (FBS, Gibco) and 1% penicillin/streptomycin (Gibco).

Wild-type bone-marrow-derived macrophage (J2-BMM) cells and interferon-α receptor-deficient (*IFNAR−/−*)-J2-BMM cells were gifted by Dr. Feng Ma (Suzhou Institute of Systems Medicine, Chinese Academy of Medical Sciences, Suzhou, China), and cultured in conditioned RPMI 1640 medium containing 10% fetal bovine serum (FBS, Gibco), 1% penicillin/streptomycin (Gibco), 10 mM HEPES (pH 7.8), and 1% M-CSF (Gibco).

The *TAP1*-deficient cell was derived from the tail skin of *TAP1*-deficient mice, which was provided by Genhong Cheng’s lab (University of California, Los Angeles, UCLA, Westwood, LA, USA).

#### 4.1.2. Viruses

HSV-GFP-Luc strain F virus was generated in our lab, and this recombinant virus was constructed by introducing a green fluorescent protein (GFP) and *Renilla* luciferase (Luc) gene into the HSV-1 genome. Ad5 was constructed and stored in our lab. VSV and SARS-CoV-2 pseudovirus was gifted by Dr. Tian Lan (VectorBuilder, Guangzhou, China). DENV4 and ZIKV were kindly gifted by Dr. Zhongyu Liu (Sun Yat-sen University, Guangzhou, China). Influenza virus A/PR/8 (H1N1) was gifted by Prof. Yuelong Shu (Sun Yat-sen University, Guangzhou, China).

### 4.2. Cell Transfection

The 80% confluence of cultured cells were transfected using Lipofectamine 2000 CD Transfection Reagent (Cat. No 12566014; Invitrogen, Carlsbad, CA, USA) with plasmids according to the manufacturer’s manuals. The medium was replaced with DMEM containing 5% FBS and 1% penicillin/streptomycin after for transfection 5 h, and then the cells were incubated for 24 h or 48 h for the expression of the specific gene.

The siRNA oligonucleotide duplexes targeting TAP1 were synthesized by Sangon (Shanghai, China). The negative-control siRNAs were purchased from Sangon. The cells were transfected with 100 nm of the indicated siRNAs for 48-72 h using Lipofectamine^TM^ RNAiMax transfection reagent (Cat. No 13778150; Invitrogen, Carlsbad, CA, USA) according to the manufacturer’s protocol. Knockdown of the target proteins was confirmed by Western blot with the indicated antibodies and quantitative real-time PCR (RT-qPCR).

### 4.3. Virus Titer

#### 4.3.1. Endpoint Dilution Assay

HSV-GFP-Luc strain F virus titer was measured by 50% tissue culture infective dose (TCID50). Briefly, Vero cells (5 × 10^3^ cells/well) were seeded in 96-well plates. After 24 h incubation, the medium was removed, and the monolayer cells were incubated for 1 h at 37 °C with serial 10-fold diluted HSV-GFP-Luc strain F virus supernatant. After 2 h of incubation, the viral inoculum was removed, and a complete DMEM culture medium (100 μL/well) was added. All plates were incubated at 37 °C in a humidified 5% CO_2_ incubator and were observed daily for cytopathic effect. The endpoints were estimated after 48 h, and titers were calculated as TCID50/mL using the Spearman-Karber method.

#### 4.3.2. Viral Plaque Assay

The Vero cells were seeded in a 12-well plate at a density of 5 × 10^5^ for 12 h, and then the virus supernatant in 10-fold serial dilution was added. After 2 h of incubation at 37 °C in 5% CO_2_, the viral inoculum was aspirated, and the cells were washed with PBS. Then, a maintenance medium containing 1% FBS, 1% low-melting-point agarose, and 1% penicillin-streptomycin was added to each well. After 5 days of incubation, the cells were fixed with paraformaldehyde for 1 h and stained with 2% crystal violet for another 1 h. Plaques were visualized and enumerated, and the virus titer in plaque-forming units per mL (PFU/mL) was calculated.

The Vero cells were transfected with plasmids (1 μg) expressing GFP, TAP1, IRF1 for 24 h, followed by infecting cells with HSV-GFP-Luc strain F virus for 8 h at MOI of 0.25, and HSV-1 titer was quantified by plaque assay.

#### 4.3.3. Luciferase Assay

The Steady-Glo^®^
*Renilla* Luciferase detection system and Dual-Luciferase Reporter Assay were employed according to the manufacturer’s instructions (Promega, Madison, WI, USA).

### 4.4. Quantitative Real Time PCR

Total RNA from HEK293T, J2-BMM or Raw 264.7 cells was extracted with TRIzol reagent (Yeasen, Shanghai, China) and reverse-transcripted to cDNA by HifairTM III 1st Strand cDNA Synthesis SuperMix for qPCR (Yeasen, Shanghai, China). The total DNA from HEK293T was extracted with a Quick Tissue/Culture Cells Genomic DNA Extraction Kit (Dongsheng Biotech, Guangzhou, China). The mRNA level and DNA level were measured using the PerfectStart SYBR Green qPCR supermix (Transgen, Beijing, China) and a QuantStudio 7 Flex quantitative real-time PCR (qPCR) detection system (Applied Biosystems, Foster City, CA, USA). Relative copies of gene expression were calculated by the comparative cycle threshold-value method, and the relative genome copy numbers were calculated based on the normalization with the housekeeping gene β-actin. The results were represented as the fold change in expression level for each transcript. The data were analyzed by the 2^−ΔΔCt^ method.

### 4.5. Western Blot Analyses

Western blot analysis was carried out by standard procedure. HSV-1 ICP-27 and influenza A neuraminidase (NA) antibodies were from Abcam (Cambridge, UK). GAPDH, TAP1, TBK1, P-TBK1 (Ser172), IRF3, P-IRF3 (Ser396), and P-STAT1 (Ser727) antibodies were all purchased from Cell Signaling Technology (CST, Danvers, MA, USA).

### 4.6. ELISA Assay

The samples were collected and analyzed with the Human IFN Beta ELISA Kit (Elabscience Biotechnology, Wuhan, China) or the Mouse IFN Beta ELISA Kit (Solarbio Life Science, Beijing, China) according to the manufacturer’s instructions.

### 4.7. Cell Viability Assay

Cells were treated with different viruses to compare the effects on cell viability. After 24 h, cell viability was determined using a Cell Counting Kit-8 (Yeasen, Shanghai, China) according to the manufacturer’s instructions. Absorbance was measured on the Synergy HTX microplate reader (BioTek, Winooski, VT, USA) at 450 nm. The results are presented as a percentage comparing the absorbance of treated samples with those of respective controls.

### 4.8. Statistical Analysis

Statistical analysis was performed using GraphPad Prism 8 software (v.8; GraphPad Software, La Jolla, CA, USA). The data are reported as mean ± SD. The unpaired *t*-test comparison was performed to assess the significant differences between two groups (* *p* < 0.05, ** *p* < 0.01, *** *p* < 0.001).

## Figures and Tables

**Figure 1 ijms-22-04668-f001:**
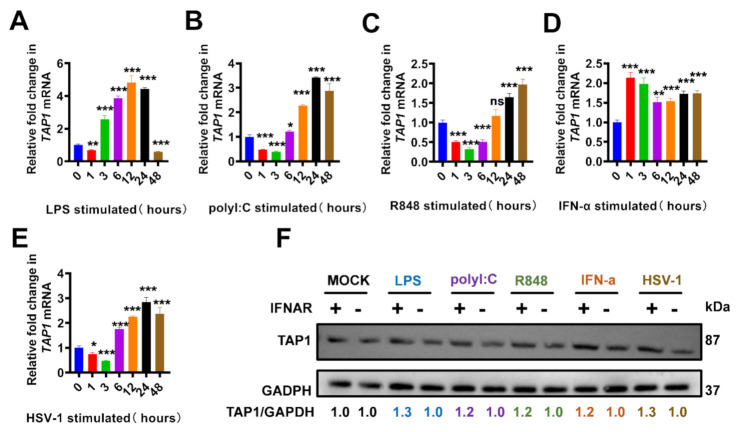
The upregulated TAP1 in response to Toll-like receptors (TLRs) agonists and viral infections. (**A**–**E**) Raw 264.7 cells were stimulated with LPS (1 μg/mL), polyI:C (25 μg/mL), R848 (100 nM), IFN-α (2000 U/mL), and HSV-1 (MOI = 0.25), respectively, and then the expression of *TAP1* gene was measured by RT-qPCR at different timepoints. The expression level of mRNA was normalized to the expression of *β-actin*, and the data from at least triplicates were shown as the mean ± SD. * *p* < 0.05, ** *p* < 0.01, *** *p* < 0.001. (**F**) Wild type bone marrow-derived macrophage (WT-J2-BMM) cells or interferon α receptor-deficient (*IFNAR−/−*)-J2-BMM cells were stimulated with LPS (1 μg/mL), polyI:C (25 μg/mL), R848 (100 nM), R848 (100 nM), IFN-α (2000 U/mL), and HSV-1(MOI = 0.25) for 24 h, respectively, and then the expression of TAP1 protein was measured by Western blot analysis. GAPDH was used as intern control. The relative ratios of TAP1 and GAPDH were marked at the bottom of the pictures.

**Figure 2 ijms-22-04668-f002:**
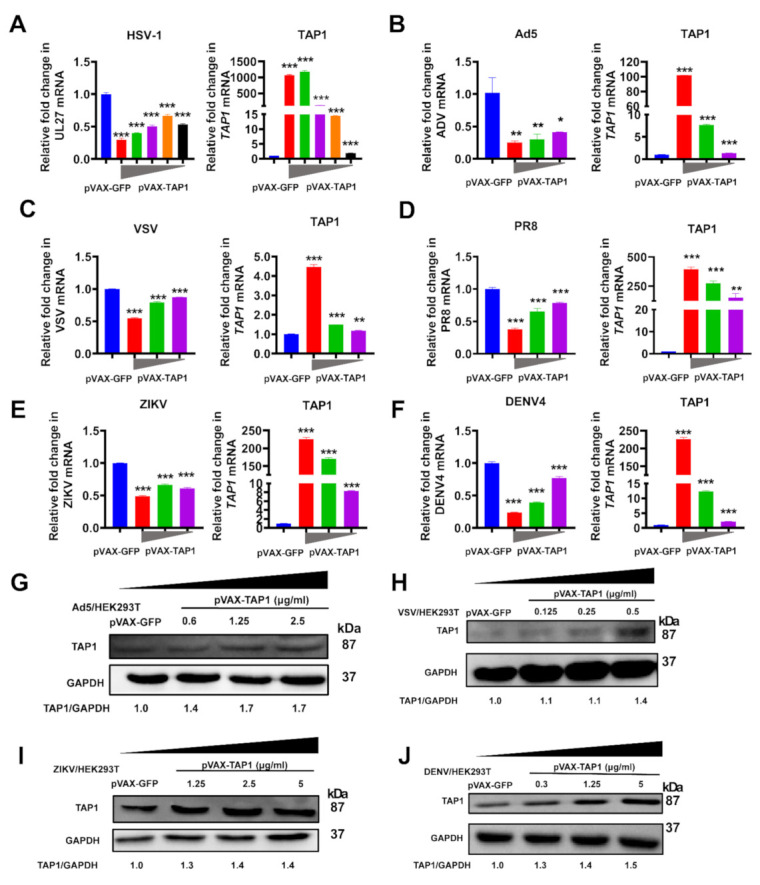
Broadly antiviral activities of TAP1. (**A**) HEK293T cells were transfected with *TAP1*-expressing plasmid in different concentrations (2.5 μg/mL, 1.25 μg/mL, 0.6 μg/mL, and 0.3 μg/mL) for 24 h, followed by HSV-1 infection (MOI = 1) for 8 h. The expression level of *TAP1* and HSV-1 *UL27* was quantified by RT-qPCR. (**B**) HEK293T cells were transfected with *TAP1*-expressing plasmid in different concentrations (2.5 μg/mL, 1.25 μg/mL, and 0.6 μg/mL) for 24 h, followed by Ad5 infection (MOI = 1) for 8 h. The expression level of *TAP1* and Ad5 was quantified by RT-qPCR. (**C**) HEK293T cells were transfected with *TAP1*-expressing plasmid in different concentrations (0.5 μg/mL, 0.25 μg/mL, and 0.125 μg/mL) for 24 h, followed by VSV infection (MOI = 1) for 8 h. The expression level of *TAP1* and VSV was quantified by RT-qPCR. (**D**) HEK293T cells were transfected with *TAP1*-expressing plasmid in different concentrations (2 μg/mL, 1 μg/mL, and 0.25 μg/mL) for 24 h, followed by PR8 infection (MOI = 1) for 8 h. The expression level of *TAP1* and PR8 was quantified by RT-qPCR (**E**) HEK293T cells were transfected with *TAP1*-expressing plasmid in different concentrations (5 μg/mL, 2.5 μg/mL, and 1.25 μg/mL) for 24 h, followed by ZIKV infection (MOI = 1) for 8 h. The expression level of *TAP1* and ZIKV was quantified by RT-qPCR. (**F**) HEK293T cells were transfected with *TAP1*-expressing plasmid in different concentrations (5 μg/mL, 1.25 μg/mL, and 0.3 μg/mL) for 24 h, followed by DENV4 infection (MOI = 1) for 8 h. The expression level of *TAP1* and DENV4 was quantified by RT-qPCR. The expression level of mRNA was normalized to the expression of *β-actin*, and data from at least triplicates were shown as the mean ± SD. * *p* < 0.05, ** *p* < 0.01, and *** *p* < 0.001. (**G**–**J**) HEK293T cells were transfected with TAP1-expressing plasmid in indicated concentrations for 24 h, followed by Ad5, VSV, ZIKV, and DENV4 infection (MOI = 1) for 8 h, respectively. The expression level of TAP1 and GAPDH was quantified by Western blot, and the relative ratios of TAP1 and GAPDH were labeled at the bottom of the pictures. GAPDH was used as the internal control.

**Figure 3 ijms-22-04668-f003:**
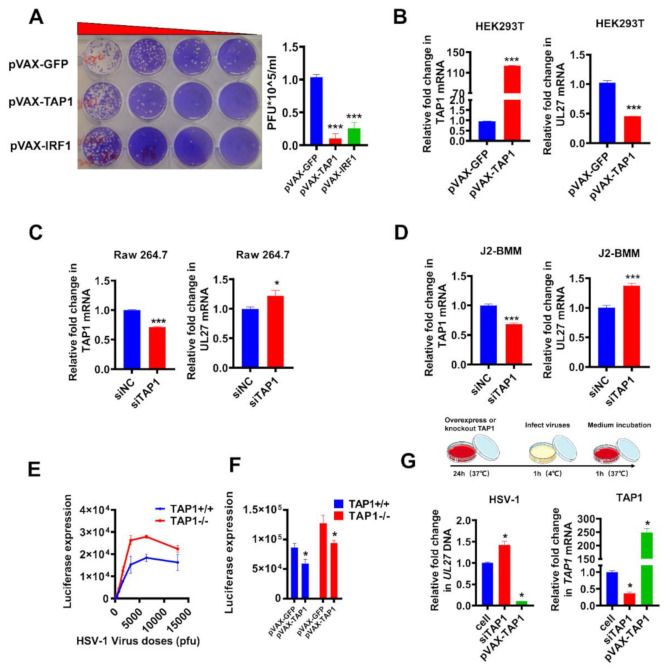
Validation of viral inhibition by loss-of-function and gain-of-function experiments. (**A**)Vero cells were transfected with plasmids (1 μg/mL) expressing *GFP*, *TAP1*, and *IRF1* for 24 h, respectively, followed by HSV-1 infection (MOI = 0.25) for 8 h. Then, the titer of HSV-1 in the supernatant was quantified by plaque assay. (**B**) HEK293T cells were transfected with plasmids (1 μg/mL) expressing *GFP* and *TAP1* for 24 h, respectively, followed by HSV-1 infection (MOI = 0.25) for 8 h. Then, the expression of *TAP1* and HSV-1 *UL27* were quantified by RT-qPCR. (**C**,**D)** Raw 264.7 and WT-J2-BMM cells were transfected with small interfering RNA (siRNA) (siNC as negative control) for 24 h, followed by HSV-1 infection (MOI = 0.25) for 8 h. Then, the expression of *TAP1* and HSV-1 *UL27* were quantified by RT-qPCR. The expression level of mRNA was normalized to the expression of *β-actin*. (**E**) *TAP1*-wildtype (*TAP1*+/+) or *TAP1*-knockout (*TAP1*−/−) fibroblast cells were infected with HSV-1, and subsequently, the Renilla luciferase activity in cell lysates was determined at 8 h post infection. (**F**) *TAP1*+/+ or T *TAP1*−/− fibroblast cells were transfected with or without *TAP1*-expressing plasmid (1 μg/mL) for 24 h, followed by HSV-1 infection (MOI = 0.25) for 8 h, and then the Renilla luciferase activity in cell lysates was determined. (**G**) HEK293 cells were transfected with plasmids (1 μg/mL) expressing *TAP1* or si*TAP1* for 24 h, respectively, followed by HSV-1 incubation (MOI = 1) for 1 h at 4 °C. Then, aspirated and discarded the supernatant, washed with PBS three times, added culture medium, incubated at 37 °C for 1 h, and the expression of HSV-1 (*UL27*) and *TAP1* were quantified by qPCR. The expression level was normalized to the expression of β-actin, and the data from at least triplicates were shown as the mean ± SD. * *p* < 0.05, and *** *p* < 0.001.

**Figure 4 ijms-22-04668-f004:**
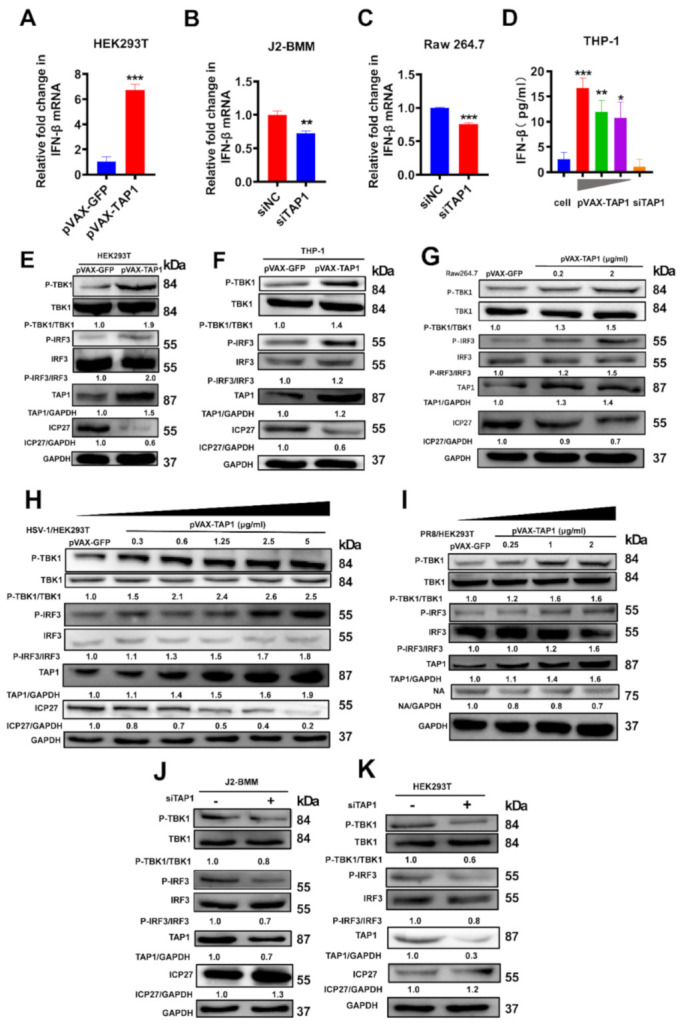
TAP1 significantly promoted the interferon (IFN)-β production through activating the TANK binding kinase-1 (TBK1) and the interferon regulatory factor 3 (IRF3) signaling transduction. (**A**) HEK293T cells were transfected with plasmids expressing *GFP* or *TAP1* (1 μg/mL) for 24 h, respectively, followed by HSV-1 infection (MOI = 0.25) for 8 h. Then, the expression of *IFN-β* was quantified by RT-qPCR. (**B**) WT-J2-BMM cells were transfected with siNC, si*TAP1* for 24 h, followed by HSV-1 infection (MOI = 0.25) for 8 h. Then, the expression of *IFN-β* was quantified by RT-qPCR. (**C**) Raw264.7 cells were transfected with siNC, si*TAP1* for 24 h, followed by HSV-1 infection (MOI = 0.25) for 8 h. Then, the expression of *IFN-β* was quantified by RT-qPCR. (**D**) THP-1 cells were transfected with *TAP1*-expressing plasmid in different concentration (5 μg/mL, 1.25 μg/mL, 0.3 μg/mL) for 24 h, followed by HSV-1 infection (MOI = 0.25). Then, the expression of IFN-β in cell lysates was determined by ELISA kits. (**E**–**G**) HEK293T, THP-1 or Raw 264.7 cells were transfected with plasmids expressing *GFP* or *TAP1* (1 μg/mL) for 24 h respectively, followed by HSV-1 infection (MOI = 0.25) for 8 h. Then, the expression of targeted proteins was detected by Western blot. The relative ratios of targeted protein and GAPDH were marked at the bottom of the pictures. (**H**,**I**) HEK293T cells were transfected with plasmids expressing *GFP* or *TAP1* for 24 h, followed by viral infections (HSV-1 or PR8) for 24 h at MOI of 1. Then, the expression of targeted proteins was detected by Western blot. The relative ratios of targeted protein and GAPDH were marked at the bottom of the pictures. (**J**,**K**) WT-J2-BMM or HEK293T cells were transfected with siNC or si*TAP1* for 24 h, followed by HSV-1 infection (MOI = 0.25) for 8 h. Then, the expression of targeted proteins was detected by Western blot. The relative ratios of targeted protein and GAPDH were marked at the bottom of the pictures. (**L**) HEK293T cells were transfected with or without *TAP1*-expressing plasmid for 24 h, followed by viral infections for 8 h at MOI of 1, including VSV, ZIKV, DENV4, Ad5 and PR8. Then, the expression of *IFN-β* was quantified by RT-qPCR. (**M**,**N**) WT-J2-BMM or Raw 264.7 cells were transfected with *TAP1*-expressing plasmid for 24 h, followed by viral infections for 24 h at 1 MOI, including HSV-1, VSV, Ad5, SARS-CoV-2 pseudovirus, ZIKV, DENV4. Then, the expression of IFN-β in cell lysates was determined by ELISA kits. (**O**) (*IFNAR−/−*)-J2-BMM cells were transfected with TAP1-expressing plasmid for 24 h, followed by HSV-1 or SARS-CoV-2 pseudovirus infection for 24 h. Then, the expression of IFN-β in cell lysates and supernatant was determined by ELISA kits. (**P**) WT-J2-BMM, (*IFNAR−/−*)-J2-BMM, and Raw 264.7 cells were transfected with *TAP1*-expressing plasmid for 24 h, followed by HSV-1 infection for 24 h. Then, the level of STAT1 phosphorylation and total STAT1 was measured by Western blot. The data from at least triplicates were shown as the mean ± SD. * *p* < 0.05, ** *p* < 0.01 and *** *p* < 0.001.

**Figure 5 ijms-22-04668-f005:**
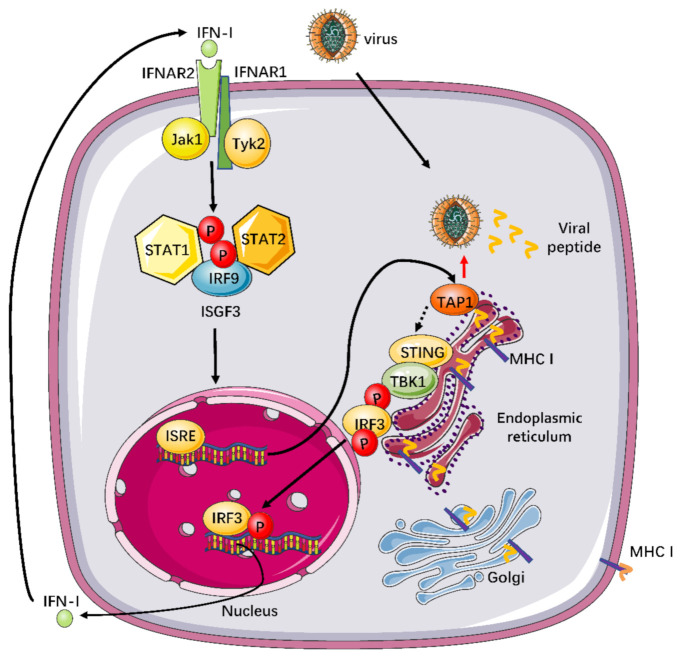
Working model to illustrate the TAP1-involved innate signal pathway against broadly viral infections. TAP1 plays a critical factor in the profound virus-host interactions, especially in innate and adaptive immunity. TAP1 can regulate adaptive immunity through promoting the major histocompatibility complex (MHC) class I-mediated antigen presentation. Additionally, our study also demonstrated that TAP1 could promote type I interferon production through activating the TBK1-IRF3 signal pathway. Subsequently, Type I IFNs can bind to IFNAR2, then recruit IFNAR1, and consequently, form a signaling-competent ternary complex to activate the transcription factor IRF9 and ISGF3 complex, which is comprised of phosphorylated STAT1 and STAT2, through the JAK-STAT signal pathway. The activated ISGF3 can translocate to the nucleus and bind to IFN-stimulated response elements (ISREs) in the upstream promoter regions of ISGs, which can encode the numerous proteins with potent antiviral activities.

## Data Availability

The data presented in this study are available in the main text and Appendix A.

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
