# Peer review of "Broadly Antiviral Activities of TAP1 through Activating the TBK1-IRF3-Mediated Type I Interferon Production"

_ijms, 2021, doi:10.3390/ijms22094668_

Round 1
Reviewer 1 Report
The manuscript by Zhao et al., confirmed TAP1 as an interferon stimulated gene and reported that TAP1 had a broad antiviral activity against HSV-1, AdV, VSV, DENV and ZIKV. The authors confirmed these observations by performing loss-of and gain-of function experiments. They found that TAP1 promoted the IFN-beta production by activating TBK1 and IRF3 signaling.
The current manuscript challenges some of the notion and previously published data on TAP1, but none of the issues were either discussed or explained in the manuscript.
- Previous studies established that TAP1 suppressed the virus-triggered induction of IFNs and ISGs or in other words knockdown of TAP1 enhances virus-triggered expression of IFNs and ISGs (PMID: 28356387). Whereas authors show that knockdown of TAP1 in BMDM or Raw264.7 cell line suppressed IFN-beta induction which is completely opposite.
- LPS and interferon can induce TAP1 expression in a STAT1 and IRF1 dependent manner has already been shown (PMID: 15240699; PMID: 12234057). Therefore, Figure 1 is redundant and should be removed.
- No explanation was provided for why Tap1 gene expression is reduced significantly before getting upregulated.
- It is very surprising that Tap1 knockdown by siRNA in bone marrow derived macrophages was achieved very easily. These cells are terminally differentiated and only lenti- or retro-viral transductions has been shown to work in some cases with 10-20% efficiency (CrispR has worked slightly better). Moreover, this is not at all mentioned in the methods section.
- The blots are of very poor quality and must be improved. Although Fig.1 is redundant, WT and IFNAR KO should have been shown side by side on the same gel for each stimulation.
- In Figure 4, to provide evidence that TAP1 mediates IFN production via activating TBK1/IRF3, overexpress TAP1 in the RAW264.7 or THP-1 cell lines and measure the phosphorylation. This can also show whether this effect is dose dependent or not.
Author Response
The manuscript by Zhao et al., confirmed TAP1 as an interferon stimulated gene and reported that TAP1 had a broad antiviral activity against HSV-1, AdV, VSV, DENV and ZIKV. The authors confirmed these observations by performing loss-of and gain-of function experiments. They found that TAP1 promoted the IFN-beta production by activating TBK1 and IRF3 signaling.
The current manuscript challenges some of the notion and previously published data on TAP1, but none of the issues were either discussed or explained in the manuscript.
Answer: Thank you for your kind comments and suggestions, and we have revised our manuscript accordingly.
- Previous studies established that TAP1 suppressed the virus-triggered induction of IFNs and ISGs or in other words knockdown of TAP1 enhances virus-triggered expression of IFNs and ISGs (PMID: 28356387). Whereas authors show that knockdown of TAP1 in BMDM or Raw264.7 cell line suppressed IFN-beta induction which is completely opposite.
Answer: Thank you for this important mention. Previously, there were several studies to report the function of TAP1, which mainly focused on host defense system and adaptive immunity. In most of these studies, TAP1 played a positive defense function against viral infection and tumorigenesis. In our work, we found that TAP1 exerted the broadly antiviral activities through activating the TBK1-IRF3 -mediated type I interferon production. Interestingly, you mentioned one study that TAP1 negatively regulated the antiviral innate immunity by targeting the TGF-β–activated kinase (TAK)1–TAK1-binding protein (TAB). The exact reason for this discrepancy is not known, but it may be due to the different experiment conditions and different innate immune signaling we studied. For example, the epithelial cell lines (A549 and Hela) were used in that study, while we used immune-related macrophages (Raw 264.7, J2-BMM, THP-1), which is more representative of the natural immune cells to study the innate immunity. More importantly, TAK1-TAB-NF-κB signaling in their study and TBK1-IRF3 signaling in our study are two different signaling pathways, and this discrepancy might be indeed a good example of the profound immune regulatory networks to maintain the immune homeostasis. We have added this discussion in revised manuscript (line 276-290). Thank you.
- LPS and interferon can induce TAP1 expression in a STAT1 and IRF1 dependent manner has already been shown (PMID: 15240699; PMID: 12234057). Therefore, Figure 1 is redundant and should be removed.
Answer: Thank you for your comment. We agree with you that It has been reported that TAP1 was induced by LPS and IFN stimulation. In our work, we initialed this study to further confirm your mentioned observation, but we are not simply duplicating the previously reported work. As shown in Figure 1, we used more different TLR agonists to stimulate different immune cell lines (J2-BMMs and IFN receptor deficient (IFNAR−/−)-J2-BMMs). Importantly, we also monitored the kinetics of TAP1 expression at multiple time-points during the stimulation, and we found an interesting observation that TAP1 expression is reduced at the early several hours upon stimulation (your next question), which is not reported previously. Therefore, we don’t think Figure 1 is redundant, and it should play a role in connecting the previous study and our next in-depth work. We also revised this Figure according your kind suggestions (refer to your question 5). Thank you for your understanding and suggestion.
- No explanation was provided for why Tap1 gene expression is reduced significantly before getting upregulated.
Answer: Thank you for your important mention. Actually, we notified that the previous work to report the increased TAP1 expression after TLR agonists stimulation was at 6-8h (PMID:28356387), or even 24-48h (PMID: 12234057, PMID:9776728, PMID:15030774), but there is no data about the early stage after stimulation. Therefore, we monitored the kinetics of TAP1 expression at multiple time-points during the stimulation, especially 1-6h after stimulation. Very interestingly, we found that there was a transient decline of TAP1 expression at the early several hours upon TLR agonists stimulation, and then followed a sharp upregulation. This is an observation that has not been reported before, and the underlying mechanism is worthy of further study in the future. We have mentioned this information in our revised manuscript (line 78-83). Thank you.
- It is very surprising that Tap1 knockdown by siRNA in bone marrow derived macrophages was achieved very easily. These cells are terminally differentiated and only lenti- or retro-viral transductions has been shown to work in some cases with 10-20% efficiency (CrispR has worked slightly better). Moreover, this is not at all mentioned in the methods section.
Answer: Thank you for letting us know your concern. We are sorry that we did not provide the detailed experimental method in the previous version for this experiment. Actually, we use J2-BMMs in our study, but not a primary BMM. BMMs were derived from mice, and J2-BMM was an immortalized cell line using J2 oncogenic retroviruses. We use Lipofectamine™ RNAiMAX Transfection Reagent (Invitrogen) for siRNA transfection, which is a unique RNAi specific cationic liposome transfection reagent, especially suitable for the delivery of siRNA and miRNA to all cell types with highest efficiency. We have added these descriptions in the methods section of the revised manuscript (line-371-374).
- The blots are of very poor quality and must be improved. Although Fig.1 is redundant, WT and IFNAR KO should have been shown side by side on the same gel for each stimulation.
Answer: Thank you for your kind suggestion. We have conducted this experiment again based on your suggestion, and the results of WT and IFNAR KO have been shown side by side on the same gel for each stimulation in the revised manuscript. Thank you.
- In Figure 4, to provide evidence that TAP1 mediates IFN production via activating TBK1/IRF3, overexpress TAP1 in the RAW264.7 or THP-1 cell lines and measure the phosphorylation. This can also show whether this effect is dose dependent or not.
Answer: Thank you for your meaningful suggestions. We have added these experiments to measure the TBK1/IRF3 phosphorylation in the THP-1 and Raw264.7 (revised Figure 4F and 4G), and the conclusion is consistent with our previous data. We also further measured this effect was dose-dependent in HEK293T cells (revised Figure 4H and 4I). We have added the corresponding descriptions in our revised manuscript (line 202-line212).
Thank you again for all of your questions and suggestions again.
Reviewer 2 Report
In this manuscript the authors seek to delineate the role of TAP1 in the innate immune response to viral infections. They find that stimulating TLR signaling and viral infection upregulated TAP1. The effect of TAP1 upregulation was mimicked by overexpression of increasing amounts of TAP1 in cells followed by infection with various viruses, which resulted in a decrease in viral RNA load. Viral loads were rescued in cells where TAP1 was knocked down, indicating a vital role for TAP1. They then show the activation of TBK1 and IRF3 during infection, which is enhanced when TAP1 is overexpressed in the cells, and reduced when TAP1 expression is knocked down. The effect on IFN-beta was also investigated and TAP1 expression correlated directly with IFN-beta production. Together, the authors try to make a case for a signaling pathway via TAP1 that regulates innate immune responses such as production of type 1 IFN and the players of the pathway, TBK1 and IRF3.
Unfortunately, the data provided is less than compelling to sufficiently make this case. In addition, the literature cited sometimes seems in conflict with the reports being made, which is not addressed. It is this reviewer’s recommendation to clarify the following points before the paper can be considered for publication:
Major concerns:
- There appears to be a discrepancy between the fold upregulation of TAP1 mRNA as estimated by RT-PCR and by western blotting. In figure 1 F, the authors claim that there is a difference in TAP1 expression on stimulation with IFN-alpha and on infection by HSV-1, but the < 3 fold increase in mRNA doesn’t seem to translate into any increases in protein levels.
- There appears to be decrease in TAP1 expression in the IFNAR-/- BMMs when treated with LPS or IFN-alpha- any explanation?
- Figure 2: there is a wide range by which TAP1 mRNA expression is increased- from 200 to 1500 fold. What accounts for this? There doesn’t seem to be correlation with the DNA transfected. Fig 2A says highest DNA concentration is 2.5 ug/ul and there is a 1000-fold increase. In Figure 2C , 5 ug/ul was the highest concentration of TAP1 but the fold increase was around 250. What was used as the base? Infected cells or uninfected? The latter should not vary between experiments. Does this reflect a change in protein levels of TAP1? Showing TAP1 levels in uninfected, mock-transfected cells, infected mock-transfected and the infected and TAP1-trasnfected cells for at least the highest and lowest amounts of DNA transfected will be ideal.
- Interaction between TBK1 has previously been shown not to occur (REF33: . Inducible TAP1 Negatively Regulates the 459 Antiviral Innate Immune Response by Targeting the TAK1 Complex ). How does this square with the premise of this paper that TAP1 induces activation of TBK1? Please at least discuss.
- Does overexpression of TAP1 have any effect on the viral entry, such as on the receptor HSV-1 uses for entry? Is the susceptibility of infection of the TAP1 knockdown cells different or is viral replication in these cells different Line 137-139 ? How do the authors differentiate this?
- In Figure 4: the knockdown of TAP1 by si-1-3 seems equally effective, but the difference in TBK1 and IRF3 phosphorylation appears to only be significant in si-3, however the increase in viral replication is seen in all 3. If IRF-3 phosphorylation barely differs between the si-1 and si-2, but there is a difference in viral production, it implies something else is permitting replication that is dependent on TAP1, but not TBK1/IRF3. Knocking down TBK1 or IRF3 may help clarify that.
- If the authors want to really pinpoint the signaling pathway, please establish how TAP1 interacts with or regulates the phosphorylation of TBK1.
Minor issues:
- Figure 4K: please show the total STAT blot
- Why did the authors choose to look at TBK1? What was the rationale? Please make it clear in the paper so the reader not intimately familiar with the field will know.
- Please ensure the GAPDH normalization is not saturated so the ratios are accurate.
- Some light language and copy editing is required.
Author Response
- There appears to be a discrepancy between the fold upregulation of TAP1 mRNA as estimated by RT-PCR and by western blotting. In figure 1 F, the authors claim that there is a difference in TAP1 expression on stimulation with IFN-alpha and on infection by HSV-1, but the < 3 fold increase in mRNA doesn’t seem to translate into any increases in protein levels.
Answer: Thank you for your careful observation. To further confirm our observation, we detected the change of TAP1 expression by both RT-PCR and Western blotting. However, as well known, there are usually some differences between gene transcription level and protein translation level. Although the change values were different to some extent, the change trend is consistent both in our previous experiment and our recent repeated experiment. Thank you for your understanding.
- There appears to be decrease in TAP1 expression in the IFNAR-/- BMMs when treated with LPS or IFN-alpha- any explanation?
Answer: Thank you for your mention. We don’t think there is a significant change in TAP1 expression in the IFNAR-/- BMMs when treated with LPS or IFN-alpha. We have repeated this experiment and the result was showed in the revised Figure 1F. Thank you.
- Figure 2: there is a wide range by which TAP1 mRNA expression is increased- from 200 to 1500 fold. What accounts for this? There doesn’t seem to be correlation with the DNA transfected. Fig 2A says highest DNA concentration is 2.5 ug/ul and there is a 1000-fold increase. In Figure 2C, 5 ug/ul was the highest concentration of TAP1 but the fold increase was around 250. What was used as the base? Infected cells or uninfected? The latter should not vary between experiments. Does this reflect a change in protein levels of TAP1? Showing TAP1 levels in uninfected, mock-transfected cells, infected mock-transfected and the infected and TAP1-trasnfected cells for at least the highest and lowest amounts of DNA transfected will be ideal.
Answer: Thank you for your careful reading and kind suggestions.
1)We also notified the wide range for TAP1 mRNA expression, one possible reason might be that the different viruses had various effects on cell viability (Figure S1), which may subsequently affect the level of TAP1 expression (line112 -line114).
2)The base for TAP expression level is these cells that was transfected with pVAX-GFP control plasmid and then infected with different viruses.
3)According to your suggestion, we have added the data of TAP1 protein level (Figure 2G-J, Figure 4H and 4I) (line109 -line112).
4)The comparison of TAP1 levels in uninfected, mock-transfected cells, infected mock-transfected and the infected and TAP1-trasnfected cells was showed in Figure S2. The mock-transfected cells didn’t affect TAP1 expression, and TAP1 expression was upregulated about 2 folds in infected mock-transfected cells. TAP1 was upregulated about 100 folds in infected TAP1-trasnfected cells (line114 -line116).
We have mentioned these descriptions in revised manuscript accordingly. Thank you.
- Interaction between TBK1 has previously been shown not to occur (REF33: . Inducible TAP1 Negatively Regulates the 459 Antiviral Innate Immune Response by Targeting the TAK1 Complex). How does this square with the premise of this paper that TAP1 induces activation of TBK1? Please at least discuss.
Answer: Thank you for your question. As you mentioned, the previous study showed that there was no direct binding between TBK1 and TAP1, while our results showed that TAP1 promoted the phosphorylation of TBK1. We speculate that TAP1 can promote the phosphorylation of TBK1 by an indirect manner, which we still do not know the exact pathway. The next work should clarify how TAP1 exactly promotes the phosphorylation of TBK1. We have added this discussion in revised manuscript (line290-294). Thank you for your suggestions.
- Does overexpression of TAP1 have any effect on the viral entry, such as on the receptor HSV-1 uses for entry? Is the susceptibility of infection of the TAP1 knockdown cells different or is viral replication in these cells different Line 137-139? How do the authors differentiate this?
Answer: Thank you for your meaningful question. We also explored whether TAP1 inhibits the entry of HSV-1. The TAP1 in HEK293T cells was overexpressed or knockdown for 24 hours, and then HSV-1 was incubated at 4°C for 1h to allow the viruses to adsorb onto the cells. After washing out the remaining viruses, the culture was incubated at 37°C for 1 hour for the HSV-1 entry into cells (but have no enough time to replicate to produce the progeny virus (Figure 3G), and then HSV-1 copies were detected. Our result showed that TAP1 overexpression inhibited the viral entry, while TAP1 knockdown promoted the viral entry (Figure 3G). However, TAP1 is an endoplasmic reticulum protein that probably does not affect extracellular HSV-1 entry directly, and the mechanism is not known yet. We have added this information in the revised manuscript (line153 -line159).
- In Figure 4: the knockdown of TAP1 by si-1-3 seems equally effective, but the difference in TBK1 and IRF3 phosphorylation appears to only be significant in si-3, however the increase in viral replication is seen in all 3. If IRF-3 phosphorylation barely differs between the si-1 and si-2, but there is a difference in viral production, it implies something else is permitting replication that is dependent on TAP1, but not TBK1/IRF3. Knocking down TBK1 or IRF3 may help clarify that.
Answer: Thank you for your careful reminders. As you mentioned, there were different effect on TBK1 and IRF3 phosphorylation by different siRNA1-3, might because these siRNAs are targeted different region in TAP1. The accurate ratio value is different for different siRNAs, but they all show a downward trend both in our previous experiment and our recent repeated experiment. In order to make this question more concise, we only show the data of siRNA3 in the revised Figure 4.
It is a good suggestion for the knockdown of TBK1 or IRF3, but we currently have no these reagents. We will perform it in our next project to clarify how TAP1 exactly promotes the phosphorylation of TBK1. Thank you for your understanding and suggestions.
Minor issues:
- Figure 4K: please show the total STAT blot
Answer: Thank you for your kind reminder. We have provided the total STAT data in the revised Figure 4P.
- Why did the authors choose to look at TBK1? What was the rationale? Please make it clear in the paper so the reader not intimately familiar with the field will know.
Answer: Thank you for your kind suggestions. To explore the underlying mechanism for broadly antiviral activity of TAP1, our work showed that TAP1 overexpression significantly induced a high level of IFN-β expression, and therefore we detected the subsequent signaling pathway involved in TAP1-induced production of type I interferon. Since previous study demonstrated that the induction of type I interferons might be mainly related to the STING-TBK1-IRF3 signaling axis [ PMID:32911481], so we focused on this signaling pathway in our next study. We have mentioned this information in the revised manuscript (line200-line202).
- Please ensure the GAPDH normalization is not saturated so the ratios are accurate.
Answer: Thank you for your kind reminder. We have check it again in our revised manuscript. Thank you.
- Some light language and copy editing is required.
Answer: The revised manuscript has been carefully edited again. Thank you.
Thank you for all of your questions and suggestions again.
Round 2
Reviewer 1 Report
Authors have added new data to the revised manuscript to strengthen the findings. But the major concern still remains.
1. When the controversial/contradictory results are being reported, it should be backed upon by solid evidences and valid arguments. Just saying that others performed it in different cells or using different reagents which have resulted in the differences is not enough. In (PMID: 28356387), authors have also used THP-1 cells. In (PMID: 15240699), authors have looked at the TAP1 up regulation at earlier time points (1 and 3 hours). I have only mentioned 1 paper but there are more findings that have reported the negative regulation by TAP1. Above all, data is data and needs to be reported if backed by appropriate evidence and discussion.
2. In REF#7-9, authors claim that the positive regulation by TAP1 has been shown. I could not find any mention of positive regulation by TAP1 in those references. Please correct the references.
3. In all the western blot data, the phosphorylation or protein induction differences are very subtle and I am wondering whether this subtle difference of 1.2 or 1.3 fold can result in huge induction of the IFN production.
4. I agree that TAK-1/NF-kB signaling is different from Viral RNA/DNA induced IFN production where most of the part is played by RIG-I/MDA-5 and goes through STING/TBK-1 axis. I brought this point because authors have performed stimulation with multiple TLR ligands where LPS follows TLR4-MYD88-TAK-1-NF-kB whereas Poly I/C follows TLR3-TRIF-TBK1-IRF3. How does this pan stimulation effect fits with the proposed mechanism?
Author Response
- When the controversial/contradictory results are being reported, it should be backed upon by solid evidences and valid arguments. Just saying that others performed it in different cells or using different reagents which have resulted in the differences is not enough. In (PMID: 28356387), authors have also used THP-1 cells. In (PMID: 15240699), authors have looked at the TAP1 up regulation at earlier time points (1 and 3 hours). I have only mentioned 1 paper but there are more findings that have reported the negative regulation by TAP1. Above all, data is data and needs to be reported if backed by appropriate evidence and discussion.
Answer: Thank you for your kind comments. We totally agree with you that “data is data and needs to be reported if backed by appropriate evidence and discussion”. In this study, we mainly investigated the TAP1-involved signal pathway against viral infections. During we performed this project, we realized that there were some discrepancy reports about TAP1 function and the following signaling transduction. For example, there were some studies to report the role of TAP1 on host defense system and adaptive immunity against viral infection and tumorigenesis. However, there were other studies showed that TAP1 might not play a role in inhibition of some viruses. For example, TAP1 did not play a significant role in inhibition of MHV-68. Moreover, some studies demonstrated that TAP1 negatively regulated the antiviral innate immunity. At present, we do not know the exact reason for this discrepancy. This discrepancy might be indeed a good example of the profound immune regulatory networks to maintain the immune homeostasis. According to your kind suggestions, we have mentioned these different data and conducted in-depth discussion in the revised manuscript (line 78-82; line 271-285). Thank you for your nice suggestions.
- In REF#7-9, authors claim that the positive regulation by TAP1 has been shown. I could not find any mention of positive regulation by TAP1 in those references. Please correct the references.
Answer: Thank you for your kind reminder. We rewrote the related description in the Introduction section (line 45-55) and Discussion section (line267-270) in revised manuscript.
- In all the western blot data, the phosphorylation or protein induction differences are very subtle and I am wondering whether this subtle difference of 1.2 or 1.3 fold can result in huge induction of the IFN production.
Answer: Thank you for letting us know your concern. We agree with you that the differences of phosphorylation or protein induction in some Figures (Figure 1F and Figure 4F) is not robust. However, we repeated these experiments many times, and these differences are stable. More importantly, these small changes in phosphorylation or protein levels in our study had indeed played a significant role in signaling transduction against various infections. One possible explanation is that the small changes might be amplified by the following cascade amplification. In addition, there is usually a profound immune regulatory network to maintain the immune homeostasis, Besides of the TBK1-IRF3 signaling in our study, the different innate signaling pathways might also be involved in TAP1-related signaling pathway to contribute to the production of IFN, and we will explore the different mechanisms in the future study. Thank you for your suggestion.
- I agree that TAK-1/NF-kB signaling is different from Viral RNA/DNA induced IFN production where most of the part is played by RIG-I/MDA-5 and goes through STING/TBK-1 axis. I brought this point because authors have performed stimulation with multiple TLR ligands where LPS follows TLR4-MYD88-TAK-1-NF-kB whereas Poly I/C follows TLR3-TRIF-TBK1-IRF3. How does this pan stimulation effect fits with the proposed mechanism?
Answer: Thank you for your kind comment and question. To further verify that TAP1 is an interferon-dependent ISG, different immune cells were stimulated with multiple TLR ligands and viral infections. As expected, TAP1 is induced by these different TLR ligands or viral infections. As you mentioned, different signaling pathway can be activated by different TLR ligands. However, our next aim is to study the possible signaling pathway by which TAP1 plays a broad antiviral activity, but not these TLR ligands-involved signaling pathway. Interestingly, as mentioned above, the different signaling pathways and functions for TAP1 had been reported. For example, there were several previous studies to report that TAP1 played defense function against viral infection and tumorigenesis, and our data supported that TAP1 exerted the broadly antiviral activities through activating the TBK1-IRF3-mediated type I interferon production. However, some study showed that TAP1 negatively regulated the antiviral innate immunity by targeting the TGF-β–activated kinase (TAK)1–TAK1-binding protein (TAB) complex. The reason for this discrepancy is not known, and it is worthy of further study in the future. Thank you for your suggestion.
Reviewer 2 Report
The authors have addressed most questions, but there is one issue still regarding Figure 4. The explanation regarding siRNA1&2 and subsequent removal of the data is still unconvincing that it is TAP that is responsible for TBK1 and IRF3 phosphorylation. The recommendation is to rescue the phenotype in the siRNA treated cells by transfecting an siRNA resistant version of the gene for TAP1 to unabigiously show it is a TAP1-dependent phenomenon and not due to any off targets.
Minor:
Put experimental details in Methods section instead of legend and the main text
Author Response
- The authors have addressed most questions, but there is one issue still regarding Figure 4. The explanation regarding siRNA1&2 and subsequent removal of the data is still unconvincing that it is TAP that is responsible for TBK1 and IRF3 phosphorylation. The recommendation is to rescue the phenotype in the siRNA treated cells by transfecting an siRNA resistant version of the gene for TAP1 to unabigiously show it is a TAP1-dependent phenomenon and not due to any off targets.
Answer: Thank you for your nice suggestion to further improve our work. According to your suggestion, we have conducted another experiment to further verify our hypothesis by rescuing the TAP1 phenotype in the siRNA treated cells. As you mentioned, it is a good idea to perform this experiment by transfecting an siRNA resistant version of TAP1 gene, but we currently cannot get this kind of TAP1 mutant gene. Alternatively, we simultaneously transfected the different concentrations of normal TAP1-expressing plasmid, when transfecting siTAP1 with the same concentration as before. The results demonstrated that TAP1 replenishment rescued the phosphorylation of TBK1/IRF3 in the siRNA treated cells in a dose-dependent manner (Figure S3, Please see the Figure S3 in the attached Word file). Taken together our previous data that the phosphorylation of TBK/IRF3 was effectively promoted by transfecting TAP1-expressing plasmid in THP-1, HEK293T and Raw264.7 cells (Figure 4E-4I), while the phosphorylation of TBK/IRF3 was effectively inhibited by transfecting siTAP1 in J2-BMM and HEK293T (Figure 4J and 4K), all of these data indicated that TAP1 affected the viral infections through regulating TBK1/IRF3 signal pathway.
- Minor: Put experimental details in Methods section instead of legend and the main text
Answer: The revised manuscript has been carefully edited according to your suggestion. Thank you again for all of your suggestions and comments to improve this work.

Round 3
Reviewer 1 Report
I am still not convinced with the explanation provided by the authors. I believe I had asked very straight forward questions which can be ruled out/explained by simple experiments with appropriate controls (positive and negative), but the description/discussion provided is not satisfactory enough to explain the findings.
Author Response
We appreciate for all of your kind questions and comments to improve our work. During the several rounds of revision, we try our best to perform a series of experiments and revise some discussions according to your helpful comments and suggestions. Now, according to both your suggestions and Editor’s suggestions, we have reorganized the relevant descriptions and made some rhetorical changes again to appropriately address concerns (Highlighted in yellow color in the revised manuscript). Scientifically, our study highlighted the broadly antiviral function of TAP1 by regulating a new signaling pathway of innate immunity, which will provide insights to develop novel strategies against emerging infectious diseases. Thank you very much for your understanding and kind comments.
Reviewer 2 Report
The authors have addressed all issues satisfactorily.
Author Response
Thank you for all of your valuable suggestions and comments to improve our work.